# Aflatoxins in Maize: Can Their Occurrence Be Effectively Managed in Africa in the Face of Climate Change and Food Insecurity?

**DOI:** 10.3390/toxins14080574

**Published:** 2022-08-22

**Authors:** Queenta Ngum Nji, Olubukola Oluranti Babalola, Mulunda Mwanza

**Affiliations:** 1Food Security and Safety Focus Area, Faculty of Natural and Agricultural Sciences, North-West University, Private Bag X2046, Mmabatho 2735, South Africa; 2Department of Animal Health, Faculty of Natural and Agricultural Sciences, North-West University, Private Bag X2046, Mmabatho 2735, South Africa

**Keywords:** aflatoxin, climate change, maize, regulation, food insecurity, predictive model

## Abstract

The dangers of population-level mycotoxin exposure have been well documented. Climate-sensitive aflatoxins (AFs) are important food hazards. The continual effects of climate change are projected to impact primary agricultural systems, and consequently food security. This will be due to a reduction in yield with a negative influence on food safety. The African climate and subsistence farming techniques favour the growth of AF-producing fungal genera particularly in maize, which is a food staple commonly associated with mycotoxin contamination. Predictive models are useful tools in the management of mycotoxin risk. Mycotoxin climate risk predictive models have been successfully developed in Australia, the USA, and Europe, but are still in their infancy in Africa. This review aims to investigate whether AFs’ occurrence in African maize can be effectively mitigated in the face of increasing climate change and food insecurity using climate risk predictive studies. A systematic search is conducted using Google Scholar. The complexities associated with the development of these prediction models vary from statistical tools such as simple regression equations to complex systems such as artificial intelligence models. Africa’s inability to simulate a climate mycotoxin risk model in the past has been attributed to insufficient climate or AF contamination data. Recently, however, advancement in technologies including artificial intelligence modelling has bridged this gap, as climate risk scenarios can now be correctly predicted from missing and unbalanced data.

## 1. Introduction

According to the Food and Agricultural Organization (FAO), globally, more than 197 million hectares of land are cultivated with maize, with a yield of 1.13 billion tons [1]. Therefore, the quality and safety assurance of maize for human and animal consumption is very important, especially with the growing concern of global food insecurity. One major quality and safety concern is the infection of maize kernels with mycotoxin-producing fungi. These are known to be climate-sensitive. Maize contamination is of global concern because of its significant role in the food and feed supply chain and its vulnerability to AF contamination [2]. Increased attention has been paid to AFs due to their role in reducing yields in agriculture, resulting in huge global economic losses [3,4], and their threat to food safety due to their highly toxigenic and carcinogenic nature [5,6,7,8]. *Aspergillus flavus* has both virulent and non-virulent strains, and under different climatic conditions may produce particular AFs, with aflatoxin B1 (AFB1) being the most carcinogenic AF [5].

Climate change (CC) has caused the alteration of fungal strain distribution and their associated mycotoxins that grow in a maize cultivation with different growing seasons. Kos et al. [9] concluded that increased AF levels in maize are mainly due to climatic extremes such as severe drought and high summer temperatures. Predicting the mycotoxin contamination of maize during its developmental stages or close to harvest allows for proper AF risk management in the industry by partners such as farmers, distributors, or feed producers. Preventive measures against AF contamination can be informed at the pre-harvest stage with field information. Post-harvest practices including prompt and proper drying methods and storage in appropriate conditions will minimize fungal growth and mycotoxin contamination. Global warming is significantly driving altered temperature distributions and extreme precipitation patterns. Agreement exists on the important role of drought, high temperatures, and extreme precipitation patterns on increased AF production in maize [10,11,12,13]. Other studies have found significant correlation between increased AF levels and insect-damaged crops [14,15]. The perception that a warmer year would automatically lead to an increased AF contamination risk has been debunked by the results obtained by Kaminiaris et al. [16], increasing the complexity of the problem. Other factors that contribute to the problem’s complexity include mycofloral profile and interactions, differences in each crop’s pathosystem, and interaction with an ever-changing climate. It is therefore essential to carry out predictive studies (using climate models with variables such as rainfall, humidity, and temperature) on the effects that CC may have on the presence of AFs in maize. This will address future uncertainties and highlight AF risk situations in order to handle escalated mycotoxin incidence in agricultural products and in the long run to ensure food safety with increasing CC. Researchers have long anticipated that predictive models would be useful tools in the management of plant pathogens and mycotoxins. The rise in mathematical forecasting models allows for the prediction of AF contamination risks and is widely used by stakeholders in the maize supply chain. The complexity of these prediction models varies from statistical tools such as simple regression equations to complex systems such as artificial intelligence models. A lot of these prediction models have been developed in the USA, Europe, and Australia. Predictive model development in Africa is not commonplace, and those that are in development are still in their infancy. This review seeks to explore existing AF contamination risk predictive models that have the potential of being extrapolated to help control AFs in maize cultivated in Africa. This may aid in the quest to develop similar novel models in Africa.

## 2. Climate Change and Aflatoxin Contamination of Maize

Climate change influences interactions among distinct mycotoxigenic species, and their toxins, produced in foods and feeds [17,18]. Countries within the temperate climatic zone that seem safe today, may become more vulnerable to the risk of disease and loss in crop production because of contamination such as changing climatic conditions [9,19]. The impact of CC on agricultural production is greatest in the tropics and subtropics, with sub-Saharan Africa showing high vulnerability to these impacts because of changing stresses and low adaptive capacity [20]. Africa is warming faster than the global rate and maize growing season temperatures are typically increasing [21]. Changes in climatic variables such as precipitation, increase in seasonal and extreme temperature events, and the intensity of droughts during maize growing seasons vary greatly and might result in changes in the yields of maize production [20]. In sub-Saharan Africa, maize is mainly cultivated in subsistence farming systems under non-irrigation conditions; therefore, reliance on rainfall increases the susceptibility of maize crops to CC effects [22]. Low yields in this region are mainly attributable to drought stress, low soil fertility, weeds, pests, diseases, low input availability, and inappropriate seeds. These conditions enable fungal growth and mycotoxin production, making sub-Saharan Africa a vulnerable region to the mycotoxin contamination of crops. The Mediterranean basin is experiencing noteworthy changes in rainfall, giving rise to drought, increased temperatures and elevated CO_2_, allowing for the occurrence of many adverse effects that influence food production and AF contamination in maize [23]. Rainfall variability and increased temperatures are the most significant variables of CC that have severe effects on agriculture, and by extension maize production. Particularly, high temperatures, greater CO_2_ concentrations, drought stress, and altered rainfall directly affect maize and *A. flavus* prevalence, favouring fungal growth, conidiation and spore dispersal, thereby affecting the growth of maize [24,25]. The recurrent and persistent occurrence of drought stimulates AF production by *A. flavus* in both pre-harvest and post-harvest conditions [11,12,13,26]. For example, in 2015, hot and dry climatic conditions contaminated 6% of maize fields in France with AFs and 69% of isolated strains were known *A. flavus* strains [27]. Similar results have been reported in African countries [12,28,29].

Fungal development and AF production in agricultural products is primarily based on temperature, moisture, soil type, and storage conditions [9,30]. These fungi colonize many crops and adapt to different environmental conditions, having specific and overlapping ecological niches [31]. Understanding the different climatic factors influencing fungal survival, development, metabolic activity, and interaction with other organisms such as host plants, is vital for deterring their overall behaviour, leading to toxin contamination [26]. In a study by Zuma-Netshiukhwi et al. [32], it was determined that a temperature rise by 1 °C or 2 °C will result in a roughly 20% to 25% decrease in grain yield as a result of CC. Kachapulula et al. [12] reported high levels of AFs in maize and groundnuts in a drier and low rainfall zone as compared to cooler and high-rainfall zones. Likewise, Sirma et al. [33] reported that crops cultivated in semiarid tropical regions were more prone to AF contamination than those in temperate regions. Indirect effects of CC on mycotoxin contamination include increased drought stress and insect damage to the plant. The phenology of the crop can be altered. The reproductive stages (germination, silking, pollen shedding, and grain filling) are sensitive stages of crop development, for this reason, the extent and gravity of drought during this stage can decrease crop yield to approximately 50% [34,35]. Chauhan et al. [36] postulated that the grain filling period is critical for agronomic practices in order to decrease the effects of drought and high temperatures on yield, and to lower the risk of AF contamination. Ding and Wang [37] reported high AF levels in groundnuts grown in regions with limited rainfall and high daily temperatures, or those exposed to heat stress, during the last month of the growing season. Overall, CC drives alterations in factors that have a critical impact on maize growth and yield including rainfall, pests, diseases and temperature [38]. These same conditions are favourable for fungal development and mycotoxin production. Table 1 presents AF contamination levels in maize found in some African nations between 2017 and 2022. All these countries have AF levels above their respective set standards with high contamination rates. This is a clear food safety concern.

## 3. Aflatoxin Regulation and Food Security in Africa

Due to the carcinogenic and toxigenic nature, including hepatic toxicity, of AFs, regulatory limits are placed on the quantity of AFs permitted in food and feed in several countries [51,52]. The intake of AF-contaminated staple foods is a serious health risk as the consumer will be exposed to the effects throughout their lives. Agriculture remains the main contributor to the livelihoods of the rural populations of developing countries. Subsistence farmers and their households consume high quantities of homegrown maize and the rest is sold to their immediate community, creating a milieu conducive to the increased risk of mycotoxin exposure. The regulation of mycotoxins in African countries is lacking, and where these regulations exist, they are typically only applied to export crops. Only fifteen African countries currently have mycotoxin regulatory standards [22]. Most subsistence farmers are not aware of mycotoxin regulations, and crops they consider useless as a result of mould infestation are mostly solely based on visual analysis, which is highly subjective. The enforcement of these regulations in an informal environment such as subsistence farming is unclear and possibly impractical. Therefore, regulatory standards in African countries will be difficult to enforce because of economic and food security reasons. For example, Ambler et al. [53] stated that farmers who report that their crops have lost quality usually do not dispose of them, but use the crop for household consumption.

The food and feed movement across the world, including mycotoxin-contaminated products, highlights the importance of global and country-specific mycotoxin occurrence surveys on foods. Regulatory standards are a barrier to business, particularly in regions with high levels of AF contamination, such as the case of Malawi, where it is only possible to export 4% of the maize produced to countries with stricter AF legal limits, such as the European Union and South Africa [54]. Senerwa et al. [55] estimated losses of millions of US dollars in the Kenyan dairy industry because of AF levels exceeding legal limits. Hence, AFs regulations hinder trade in these countries as contamination levels are often above legal regulatory levels (Table 1). As mentioned before, mycotoxin regulations are most enforced in African nations in crops destined for export purposes; however, many African traders are mainly concerned with domestic and regional trades rather than exports [56]. Since maize quality is reduced by mycotoxin contamination, its monetary value will be diminished because maize that was destined for food will now be degraded and directed toward feed. Hence, the anticipated financial returns based on the quantity of maize produced cannot be fulfilled due to mycotoxin contamination. This low return on investment will negatively affect the livelihood of the seller, resulting in poverty.

Food security is a serious global issue topping the development agendas of most countries, especially those in Africa. The prevalence of severe food insecurity in sub-Saharan Africa is common. For instance, one in four households in sub-Saharan Africa cannot access adequate food [57]. This worsening of the food security of this region has been attributed to climate shocks, conflicts, and economic slowdowns [1,58]. Millions of Africans could be stripped of their food supply if mycotoxin regulations were effectively enforced [28]. It has been projected that, by 2027, maize consumption will increase by 16%, especially in sub-Saharan Africa where human and livestock populations are growing rapidly. Whether this growth will increase maize exposure to mycotoxins is a matter of ongoing research [59]. This increase in maize consumption will increase the demand for maize. Factors such as CC and farming systems could directly influence mycotoxin contamination of maize. If the 2027 projections hold true, if conscious decisions are not made now to control the mycotoxin contamination of crops, for example, through substituted irrigation, the use of fungicides, and improvements in storage facilities, among others, mycotoxin contamination risk will keep on increasing, resulting in severe food insecurity.

Hoffmann and Moser [60] showed that products with a higher price are less contaminated than products sold at a lower price. Thus, in the face of food insecurity, food safety measures should not be disregarded. When managing AF levels in foods, food producers a charge higher price than other firms without this management action. Ayyat et al. [61] concluded that, when feed is treated with AF-absorbent materials, the treatment reduces toxicity in Nile tilapia, resulting in increased body mass and higher monetary value. The pertinent question is: in the current atmosphere of food insecurity, how many people are willing to pay higher prices for food where cheaper substitutes exist? The lack of awareness and farmers’ experiences are considered to be the underlying factors that contribute to their unwillingness to pay for AflaSafe food [62]. Studies carried out in centres of AF endemism in Africa showed that close to 90% of the population understood that mould poses a risk to human health. Few, however, understood what that risk is, and half believed that any toxins would be destroyed by cooking. It is also common practice for farmers who report crops as damaged by AFs to redirect the crop from the market to their personal consumption [53,63,64].

## 4. Aflatoxin Predictive Models in Africa

Since AF contamination occurs at different stages (pre-harvest and post-harvest) of the food production chain, control measures are based on these contamination stages. Some of these preventative techniques are knowingly or unknowingly implemented by subsistence farmers in Africa, either to reduce the effects of mycotoxins or as routine agronomic practices at the different crop growth stages. Proper disposal methods of aflatoxin-contaminated feed and crops is uncommon, even though the East African Community policy advises that incineration or the burial of contaminated crops or feed be practiced [65]. Care has to be taken because previously infected residues can be re-introduced into the system if not buried properly. A lot of research has concentrated on the pre-harvest and post-harvest control of AF contamination in crops, especially in Africa. Despite all the existing mitigation methods available, AF contamination still continues to be a global food safety issue, with high incidences continually being reported in Africa. Climate change remains the primary factor that drives altered fungal proliferation and mycotoxin contamination [22]. The climate is rapidly changing, which makes it more difficult to rely on mycotoxin research data of a particular season due to high interseasonal variability. Anticipatory studies at this point in time seem promising for addressing and highlighting AF risk situations on a regional basis within the African continent in the face of CC.

Mycotoxin contamination risk predictive models, incorporating AF field and climate data, will offer future solutions. Based on model application techniques, mycotoxin predictive models may be grouped as follows: mechanistic, empirical, or hybrid. Mechanistic models replicate the fundamental systems of crop and fungal developmental stages. Such models require an advanced understanding of each living system and substantial experimental research under different environments to obtain the needed input data, including temperature, rainfall, and soil properties [23]. Empirical modelling, on the other hand, uses mathematical functions to explain field conditions and the connected response variables [23]. Such a model will identify the particular environment, weather, and seasons that was involved in the model development and will require recalibration when applied in another area. Empirical models are used for conditions in an area represented by observational data. Thus, these models are unable to forecast situations that never occurred in the model development dataset, such as extreme weather events as a result of CC or new cultivation techniques. Hybrid models apply the principles of both mechanistic and empirical models. Since there are always speculations made about biophysical information and some amount of statistical analyses applied, all models are, to some extent, hybrid in nature. Therefore, their categorization is mainly based on their primary method of prediction. Keller et al. [66] proposed a hybrid model for AF prediction risk to include the advantages of both empirical and mechanistic modelling by the extension of these models to different spatial and temporal domains. Cross-validation of the three modelling methods will be useful to better comprehend the merits and drawbacks associated with each, and to help come up with a simplified model that can easily be used in the prediction of mycotoxin contamination in a given crop.

Recently, modelling techniques have been upgraded. For instance, machine learning algorithms have lately been introduced in food safety domains [67,68]. Such techniques can learn from data inputs and make data-driven forecasts. One such machine learning algorithm is Bayesian network (BN) modelling, which has been used to predict mycotoxin contamination in cereals in Serbia [69]. Bayesian network models can blend statistical relationships and expert knowledge. A BN model is a probabilistic model that is based on Bayesian statistics and decision theory in addition to graph theory. BN models can deal well with irregular data [69]. This gives BN modelling an edge over other models in predicting mycotoxins because it allows the model to run in the early maize growing period, when details of the entire growth period are unavailable. It is, therefore, useful for early warning purposes. Unlike linear regression models, BN models effortlessly analyse dependencies between variables: they manage non-linear relationships and blend numerous types of data, such as measurement data, expert skill, and consumer feedback [70]. BN models can include expert’s skill, and are pliable in adding new data to the prediction process. Additionally, BN models can provide results with incomplete data on the model input variables, with the caveat that this may influence the accuracy of the outputs. On the other hand, a simple logistic regression equation could provide accuracy of above 60% in the prediction of AF concentration in crops from a particular region [71]. Logistic regression is an advanced modelling approach that has been applied to numerous research fields, including food safety. Logistic regression estimates the parameters of the log odds of the probability of a binary event (e.g., the presence or absence of mycotoxins) [72]. BN models can be contrarily formulated without the speculations of linearity in logit or additivity [69]. This method is highly data-dependent, and the data cannot be used for agricultural conditions other than those introduced in the model development process before actual validation.

The APSIM model, a hybrid model which was first developed in Australia, has the potential of being used in Africa as the authors are extending their research to Kenya [36,73]. Peanut was used as the substrate for cardinal temperatures and, therefore, it needs to be verified for maize to increase its accuracy. The design of a practical prediction model for pre-harvest AF contamination from the APSIM model is a challenge because of its inability to reconcile the water activity parameter in field conditions [73]. The AFLA-maize model, a mechanistic model, was developed on maize grown in Italy [74]. It has been successful in predicting aflatoxin contamination in crops and is very adaptable for use in other regions and crops. The ability to assess the impact of climate change on mycotoxin risk is not restricted to Europe [74], but is being extended to other areas. Recently, the AFLA-maize model was effectively adapted to predict AFB_1_ occurrence in maize in Malawi [75]. In Greece, the AFLA-maize model was replicated in another pathosystem known as AFLA-pistachio. The AFLA-maize model is “based on two sub-models, one accounting for the host crop phenology and the other for the *A. flavus* infection cycle”. Each has their own associated advantages and disadvantages and requires different degrees of calibration and validation.

Table 2 presents different models that have been developed for mycotoxin prediction in crops, most of these models were developed in the USA, Asia and Europe.

## 5. Conclusions

The rationale behind the present review was to evaluate if AFs’ contamination of maize can be controlled or monitored in Africa in the face of CC and food insecurity. From literature, the AFLA-maize model is appropriate, since the same pathosystem i.e., the *A. flavus* infection cycle and maize phenology is being dealt with, albeit in a different geographic location. The maize plant phenology in Africa differs from those in Europe, where the model was initially developed, because of factors such as degree of growth, maize variety, prevailing weather conditions, and the employment of different farming techniques. Hence, the need for recalibration in a new location. High levels of interaction between agricultural practices complicate the undertaking of developing mathematical functions to be included in the creation of a predictive model [77]. With advancements in technology, other machine learning algorithm models or well-designed simple classic logistic regression can be used on African soil. Since all of the different modelling procedures have advantages and disadvantages, a single model that blends all the model types could be a possible solution. This will merge the models in a distinctive way and this will strengthen their merits.

## 6. Methodology

A literature review was conducted, using PRISMA (Preferred Reporting Items for Systematic Reviews and Meta-Analyses) guidelines [84] to gather information on the contamination of maize, foods, and feeds with mycotoxins in Southern Africa. A literature search was performed, using Google Scholar and key words and phrases used to extract peer-reviewed studies on mycotoxin predictive models. Key words and phrases used to access the information were: mycotoxin; aflatoxin; maize; model; prevention; and cereals. Sixty-nine articles with information related to this review were downloaded and evaluated.

## Figures and Tables

**Table 1 toxins-14-00574-t001:** Mycotoxin contamination of maize in some African nations over a 5-year period.

Country	Mycotoxin	Contamination Levels (ppb)	Contamination Rate (%)	Regulatory Limit (ppb)	References
Burkina Faso	AF	0.93-59	70	20	[39]
Burundi	AF	LOD-117	100	10	[40]
Cameroon	AFB_1_	6-645	22	20	[41]
Côte d’Ivoire	AF	30-91	96	20	[42]
Ghana	AFB_1_	0.38-348	80	15	[43]
Kenya	AFB_1_	1.69-403	97	5	[44]
Malawi	AF	0.7-140	21	3	[45]
Nigeria	AF	LOD955	57	20	[46]
Rwanda	AFB_1_	2-52	66	5	[47]
South Africa	AF	LOD-1082	28	5	[29]
Tanzania	AF	LOD-162	80	10	[48]
Togo	AF	0.17-1600	100	20	[49]
Uganda	AF	22.2-268	74	10	[44]
Zambia	AF	LOD-7408	73	10	[12]
Zimbabwe	AF	LOD-1369	52	10	[50]

**Table 2 toxins-14-00574-t002:** Aflatoxin predictive risk models in maize with the potential of application in Africa.

Name of Model	Type of Model	Country	Crop	Sample Size	Accuracy	Reference
APSIM	Hybrid	Australia	Maize	1379	69%	[36]
APSIM	Hybrid	Kenya/Australia	Maize	N/A	N/A	[73]
Maxent2	Empirical	Kenya and Mali	Any crop	N/A	N/A	[76]
AFLA-maize	Mechanistic	Italy	Maize	352	68%	[74]
AFLA-maize	Mechanistic	Europe	Maize	N/A	Future projection	[77]
Stacked gaussian	Empirical	USA	Maize	N/A	Quantified uncertainty	[78]
Multi-level modelling	Empirical	Kenya	Maize	2466	Not predictive	[79]
AFLA-pistachio	Mechanistic	Greece	Pistachio	130	80%	[16]
AFLA-maize	Mechanistic	Malawi	Maize	N/A	Future Projection	[75]
PREMA	Mechanistic and Bayesian Network	Serbia	Maize	867	83	[69]
Spatial Poisson profile regression	Empirical	USA	Maize	45 counties	Not a predictive model	[80]
Drought index (ARID)	Empirical	USA	Maize	N/A	82%	[81]
Risk in storage	Empirical	China	Maize	28	93.3%	[82]
AFLA-maize + carryover	Mechanistic	Ukraine and Netherlands	Maize	N/A	Future projection	[83]

N/A = Not applicable.

## Data Availability

Not applicable.

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
