# Peer review of "Aflatoxins in Maize: Can Their Occurrence Be Effectively Managed in Africa in the Face of Climate Change and Food Insecurity?"

_toxins, 2022, doi:10.3390/toxins14080574_

Round 1

Reviewer 1 Report

Toxins paper #1824127

Title: Aflatoxins in maize; can their occurrence be effectively mitigated in Africa in the face of climate change and food insecurity?

General impression:

This paper is a minireview aimed at investigating whether aflatoxin occurrence in African maize could be effectively mitigated in the face of increasing climate change and food insecurity using climate risk predictive models. The methodology consisted of searching current literature to gather information on mycotoxins predictive models as well as contamination of maize, foods, and feeds with mycotoxins in South Africa. The concern is that most problematic areas of Africa with high/increasing mycotoxin contamination is sub-Saharan Africa and not as much South Africa.  The study needs to include data from other parts of Africa to have a credible input for predictive modeling to be convincing.  Additionally, certain statements need to be supported by references.  Although the paper is relatively well written, it would benefit from some stylistic improvements suggested in the next section.

Specific suggestions:

Line 9-10: should read…of AF-producing fungal genera particularly in maize, which is a food staple commonly associated with mycotoxin contamination.

Line 11: should read…the management of mycotoxin risk.

Line 13: should read…in African maize could…

Line 18: should read…climate mycotoxin risk model in the past, was attributed to insufficient climate…

Line 30: should read…land are cultivated…

Line 33: should read…infection of maize kernels with…

Line 35-37: should read…because of its significant role in the food and feed supply chain and its vulnerability to AF contamination [2]. Increased attention has been paid to AFs due to their role in reducing yields in agriculture, resulting in huge global economic…

Line 39-40: should read…both the virulent and non-virulent strains, and under different climatic conditions may produce particular AFs, with aflatoxin B1…

Line 42: should read…alteration of fungal strain distribution, and their…

Line 44: should read… maize were mainly due to climatic extremes such as…

Line 47: should read… industry by farmers, collectors, or feed producers. Preventive…

(Who are “collectors”?)

Line 49: should read… practices including prompt and…

Line 55-56: should read… The perception that a warmer year would automatically lead to an increase in AF contamination risk…

Line 57: should read… al. [16], increasing…

Line 58: should read… complexity include mycofloral…

Line 61: should read… on the effects that CC may have on the presence of AFs in maize.

Line 64: should read… with increasing CC. Researchers have long anticipated that predictive models would be…

Line 65-66: should read… The rise in mathematical forecasting models allows for the prediction of AF contamination risks and is widely used…

Line 73: should read… to help control AFs in maize cultivated in Africa.

Line 79: should read… The impact of climate change on agricultural production is…

Line 82-83: This sentence needs a reference… Africa is warming faster than the global rate and maize growing season temperatures are typically increasing.

Line 101: should read… in 2015, hot…

Line 102: not clear!! Please rephrase or clarify…69 strains were isolated or 69% of isolated strains were known A. flavus strains?

Line 104: should read…AF production…

Line 109: should read…is vital for deterring their overall behavior leading…

Line 110-111: …a temperature rise by 1°C or 110 2°C will result in a roughly 20% to 25% decrease in grain yield. Due to aflatoxins??

Line 114: should read…were more prone to AF contamination than those in…

Line 119: should read…crop to approximately 50%...

Line 123: should read…or those exposed…

Line 125: should read…and yield including rainfall, pests,…

Line 127: should read…between 2017…

Line 143: should read…infestation is most often based on visual…

Line 147-148: should read…crops have lost quality, usually don’t dispose of it, but used the crop for their household consumption.

Line 167-168: should read…Africa is common. For instance, one in four households in sub-Saharan Africa cannot access adequate food. This sentence needs a reference!!

Line 182-183: This sentence is awkward and unclear.  Please clarify. What do you mean by value??

Line 183-184: should read…concluded that when feed is treated with AF absorbent materials the treatment reduces toxicity in Nile tilapia, resulting…

Line 197: should read…these preventive techniques are knowingly or unknowingly…

Line 205: should read…mitigation methods available, AF contamination…

Line 210: should read…seem promising…

Line 217: should read… different environments to obtain…

Line 232: should read…better comprehend the merits and drawbacks…

Line 234: should read…in a given crop.

Line 262: should read…used as the substrate…

Author Response

All responses have been provided

Reviewer 2 Report

The article is well structured, well written and relevant literature has been cited. The first three chapters are clearly presents state of the art and I have no remarks on that part. The fourth chapter is interesting and well structured, but nevertheless I would add something about the use of modeling and how it could be used in control and prevention of aflatoxin contamination. Clearly modeling by itself does not solve the problem, and for the reader with less experience in aflatoxins it should be shortly explained. Maybe also the title of this chapter could be changed. The chapter five, Conclusions, does not offer the answer to the question posed in the title. I suggest to enlarge a bit the chapter and explain how an appropriate modeling technique could improve aflatoxin control in Africa. In my opinion the article is good, but I suggest this minor changes before publishing.

Author Response

All issues have been addressed

Reviewer 3 Report

The manuscript deals with a very up-to-date matter in terms of increased mycotoxin-related food insecurity as a result of the predicted climate changes. 

The manuscript is well written and organized, and is in my opinion adequate for publication after a few minor adjustments.

In face of the central subject of the paper, in my opinion the title is not adequate. The title refers to mitigation, whereas the work is not describing mitigation strategies. I suggest that the word "mitigated" in the title should be substituted by "predicted".

In section 2, lines 101-103, examples should be given for one or more african countries, and not for France. I understand that this is intended to be a general example to ilustrate the idea, but african examples exist and would be more adequate.

In Table 1, correct Bukina Faso to Burkina Faso. Also, the introduction of the median (or average, if the median is not available) of AF concentration would be interesting and would properly complement the presented AF range.

In section 3, lines 140-141, please report to Table 1 as a way of clarifying the list of 15 countries. 

In section 4, the title is misleading. The section describes AF risk predictive models, not control strategies, although we might consider that predicting can be the first step towards controlling. I suggest the title "Aflatoxins predictive models in Africa " or something of the kind.

In table 2, the PREFUM model is not for Aflatoxins and should be removed.

In section 6, line 299, "South Africa" (name of the country) should be substituted by "Southern Africa countries"  (region)

Author Response

The changes have been addressed

Reviewer 4 Report

The part regarding regulation of mycotoxins in african contries should be more detailed (Chapter 3).

No information are reported about the existance of official control plants in Africa regarding mycotoxins in food. Please add some informations

Author Response

Reasons have been explained
